# MED28 Over-Expression Shortens the Cell Cycle and Induces Genomic Instability

**DOI:** 10.3390/ijms20071746

**Published:** 2019-04-09

**Authors:** Jin Gu Cho, Joon-Seok Choi, Jae-Ho Lee, Min-Guk Cho, Hong-Sook Kim, Hee-Dong Noh, Key-Hwan Lim, Byoungjun Park, Jin-Ock Kim, Sang Gyu Park

**Affiliations:** 1Department of Pharmacy, College of Pharmacy, Ajou University, 206 World Cup-ro, Yeongtong-gu, Suwon-si, Gyeonggi-do 16499, Korea; jgcho84@gmail.com (J.G.C.); hong2kim2@gmail.com (H.-S.K.); nohedong@naver.com (H.-D.N.); limkhwan@gmail.com (K.-H.L.); kjo8909@ajou.ac.kr (J.-O.K.); 2College of Pharmacy, Daegu Catholic University, Hayang-ro 13-13, Hayang-eup, Gyeongsan-si, Gyeongbuk 38430, Korea; joonschoi@naver.com; 3Department of Biochemistry and Molecular Biology, Ajou University School of Medicine, Suwon 16499, Korea; jhlee64@ajou.ac.kr (J.-H.L.); broselack@naver.com (M.-G.C.); 4Genomic Instability Research Center, Ajou University School of Medicine, Suwon 16499, Korea; 5Department of Biomedical Science, Graduate School of Ajou University, Suwon 16499, Korea; 6Department of Molecular Biology, Dankook University, Cheonan-si, Chungnam 31116, Korea; bbj0405@naver.com

**Keywords:** aneuploidy, cell cycle, transcription factor, micronucleus, nuclear budding

## Abstract

The mammalian mediator complex subunit 28 (MED28) is overexpressed in a variety of cancers and it regulates cell migration/invasion and epithelial-mesenchymal transition. However, transcription factors that increase MED28 expression have not yet been identified. In this study, we performed a luciferase reporter assay to identify and characterize the prospective transcription factors, namely E2F transcription factor 1, nuclear respiratory factor 1, E-26 transforming sequence 1, and CCAAT/enhancer-binding protein β, which increased MED28 expression. In addition, the release from the arrest at the G1−S or G2−M phase transition after cell cycle synchronization using thymidine or nocodazole, respectively, showed enhanced MED28 expression at the G1−S transition and mitosis. Furthermore, the overexpression of MED28 significantly decreased the duration of interphase and mitosis. Conversely, a knockdown of MED28 using si-RNA increased the duration of interphase and mitosis. Of note, the overexpression of MED28 significantly increased micronucleus and nuclear budding in HeLa cells. In addition, flow cytometry and fluorescence microscopy analyses showed that the overexpression of MED28 significantly increased aneuploid cells. Taken together, these results suggest that MED28 expression is increased by oncogenic transcription factors and its overexpression disturbs the cell cycle, which results in genomic instability and aneuploidy.

## 1. Introduction

The mediator complex is known to serve as an intermediary between transcription factors and RNA polymerase II (pol II) for the transmission of intracellular signals. The mediator complex can be divided into four distinct modules: a head module, a middle module, and a tail module, which function as the main components, and the CDK8 kinase module, which is transiently associated with the complex [1,2]. After the recruitment to DNA through the mediator-transcription complex, the mediator complex controls a variety of processes that are critical for transcription, including the reorganization of chromatin architecture and the regulation of pol II-mediated preinitiation, initiation, and elongation [3] in transcription. When considering that the essential role of the mediator complex in transcriptional control, genetic variation, or change in the expression of the mediator complex subunits that affect the pol II activity could lead to a variety of diseases. For instance, mutations in MED12, MED17, and MED23 are associated with X-linked mental retardation syndrome, infantile cerebral atrophy, and intellectual disability, respectively [2,4,5,6,7]. Mutational analysis of the mediator complex subunits can provide valuable insights that could be useful in the development of novel drugs against critical human diseases [8].

Mammalian mediator complex subunit 28 (MED28) belongs to the “head” module of the mediator complex and it was first identified as an endothelial-derived gene 1 using tumor-conditioned media of endothelial cells [9]. MED28 is highly expressed in breast, colon, and prostate cancers, and MED28 induces cancer cell proliferation via the mitogen-activated protein kinase kinase-1 (MAP2K1) signaling pathway [10,11,12]. Recently, we reported that the interplay between MED28 and ZNF224, a Krüppel-associated-box-containing zinc finger protein transcriptional cofactor, was associated with cancer cell proliferation upon initiation of the DNA damage response [13]. A study in knockout mice has shown that the genetic deletion of *MED28* results in peri-implantation embryonic lethality by reducing the expression of OCT4 and NANOG, which are pluripotency transcription factors [14]. Although the expression level of MED28 is closely associated with cell proliferation, the regulatory mechanism that is involved in enhancing MED28 expression is unknown.

Chromosome segregation is the most critical event in the cell cycle, and chromosome mis-segregation can be observed by the direct examination of chromosome movements. A high degree of mis-segregation is called chromosomal instability, and the persistent mis-segregation of chromosomes at a high rate causes aneuploidy in tumors with chromosome numbers in the range of 40–60 [15]. Aneuploidy is caused by various factors, including chemicals, environmental toxins, and DNA replication errors, and it induces increased proliferation with an abnormal cell cycle [16,17]. It is known that the alteration of the cell cycle by aneuploidy can change the intracellular or extracellular environments, thereby inducing resistance to chemotherapeutic drugs [18].

Although the molecular mechanisms underlying MED28-mediated oncogenesis are unknown, previous studies have suggested that MED28 can increase cancer cell proliferation, and phenotypes that are related to the dysregulation of MED28 have been demonstrated in breast cancer cells [11,13,19,20]. In this study, we aimed to identify and characterize the transcription factors that increase MED28 expression and investigated the involvement of MED28 in cell cycle regulation.

## 2. Results

### 2.1. Identification of the Transcription Factors

To identify the promoter region of MED28, we cloned a −3.0 kb region upstream of a putative *MED28* transcription start site and performed deletion mapping analysis. However, there was no difference in the luciferase activity until the −0.5 kb region (results not shown). We then constructed serial deletion mutants down to position −0.1 kb and observed that there was still no difference in the luciferase activity among the constructs (Figure 1A). Therefore, we analyzed the −0.1 kb promoter region for putative transcription factor–binding sites on the gene-regulation.com website. We found putative binding sites for transcription factors, including E2F transcription factor 1 (E2F-1; −44 to −37 bp), nuclear respiratory factor 1 (NRF-1; −39 to −28 bp), E-26 transforming sequence 1 (ETS-1; −43 to −37 bp and −10 to −3 bp), and CCAAT/enhancer-binding protein β (C/EBPβ; −16 to −13 bp and −6 to −3 bp; Figure 1B). Mutant promoters were generated and luciferase activity was examined to further confirm the binding site of the transcription factors. As depicted in Figure 1B, all of the mutants manifested significantly reduced luciferase activity when compared to the luciferase activity in the wild-type promoter, suggesting that transcription factors could bind to the indicated region of the MED28 promoter. Furthermore, we evaluated whether E2F-1, NRF-1, ETS-1, and C/EBPβ could affect the transcription of *MED28*. We measured the luciferase activity of the transcription factors after the transfection of DNA encoding the indicated transcription factor into HEK293 cells. As shown in Figure 1C, the transcription factors significantly increased luciferase activity when compared to the luciferase activity of the empty vector (E.V.). In addition, western blot analyses confirmed that the overexpression of E2F-1, NRF-1, ETS-1, and C/EBPβ increased MED28 expression, which is in line with the results of the luciferase assay (Figure 1D–G). To further confirm the specific binding of these transcription factors to putative binding elements of the MED28 promoter, a chromatin immunoprecipitation (ChIP) assay was performed. As illustrated in Figure 1H, the transcription factors, including E2F-1, NRF-1, ETS-1, and C/EBPβ, successfully bound to the MED28 promoter. Afterwards, we examined whether these four screened transcription factors could have synergistic effect on MED28 expression. The cotransfection of E2F-1 with C/EBPβ, NRF-1 with C/EBPβ, ETS-1 with E2F-1, or ETS-1 with C/EBPβ synergistically enhanced MED28 expression, but not after the co-transfection of NRF-1 with E2F-1 or NRF-1 with ETS-1 (Figure 1I). Taken together, these results indicated that the four transcription factors increase MED28 expression at the transcriptional and translational levels.

### 2.2. MED28 Expression is Increased at the G1−S and G2−M Transitions of the Cell Cycle

E2F-1 is a well-known transcription factor that predominantly binds to target genes that are induced at the G1–S phase transition or during the DNA damage response [21]. Similarly, ETS-1 is involved in the G1–S phase transition through the transcriptional upregulation of both *cyclin E* and *CDK2* genes [22]. Moreover, NRF-1 and C/EBPβ are also known to be associated with cell cycle regulation [22,23]. Therefore, we hypothesized that the expression pattern of MED28 may change during the cell cycle. We synchronized HeLa cells at the boundary of the G1–S transition using a double thymidine block to evaluate the changes in the expression of MED28 during the cell cycle. After the release of the cells from the G1–S transition block, the cells were harvested at various time points, as indicated and analyzed by western blotting and flow cytometry. Of note, the protein expression of MED28 gradually increased from hour 0 to hour 4 after the release, as the cells progressed through the G1–S transition, and then decreased from hour 6 to hour 10, as the cells progressed from the mid-S to mid- G2 phase (Figure 2A,B, Appendix A). In addition, we observed an increased expression level of MED28 from the late G2 to M phase, and then its expression decreased. After that, to further confirm the changes in MED28 expression, the HeLa cells were arrested at the G2−M transition using thymidine-nocodazole. The synchronized HeLa cells were released from the G2−M arrest and then harvested at various time points, as indicated. The analyses using immunoblot and flow cytometry showed that the protein expression of MED28 was increased up to hour 2 (M phase), decreased at 4 h (early G1 phase), and then increased up to hour 8 (late G1 phase) after the release from the nocodazole block (Figure 2C,D, Appendix A). These results suggest that MED28 might be involved in the regulation of the G1–S and G2–M transitions.

### 2.3. Overexpression of MED28 Induces Aberrant Cell Cycle and Aneuploidy

To investigate the function of MED28 overexpression at G1–S and G2–M transitions, we transfected MED28 into the HeLa cells and selected stable empty-vector or MED28-expressing cell lines with G418 (Figure 3A). Cell cycle analyses using a live cell imaging system showed that the average time point of cells with the empty-vector and Myc-MED28 was 18.3 ± 1.7 h (mean ± SD; interphase, 17.7 ± 1.7 h; mitosis, 0.7 ± 0.1 h,) and 14.0 ± 1.7 h (mean ± SD; interphase, 13.6 ± 1.8 h; mitosis, 0.4 ± 0.1 h), respectively, suggesting that the overexpression of MED28 significantly shortened the cell cycle in both interphase and mitosis (Figure 3B). Notably, the over-expression of MED28 significantly reduced the duration of anaphase onset when compared to the empty-vector control (empty-vector, 37.6 ± 7.8 min; Myc-MED28, 21.0 ± 3.6 min; Figure 3C). Furthermore, the overexpression of MED28 promoted chromosome condensation, segregation, cytokinesis, and decondensation, thus leading to the significant shortening of the mitotic phase (Figure 3D).

Next, we investigated the effect of the overexpression of MED28 on phases of the cell cycle. Fluorescence-activated cell sorting (FACS) analyses indicated that the overexpression of MED28 in HeLa and MCF7 cells decreased the populations of cells in the G1 and S phases and increased the population of cells at the G2−M transition. The overexpression of MED28 also shortened the cell cycle in the MCF7 cells (Figure 4A–E, Appendix A). Moreover, MED28 overexpression increased the cell population, exhibiting polyploidy of over 4n (tetraploidy), which may be a population that is aneuploid due to chromosomal instability (Figure 4A,B, Appendix A). In addition, the knockdown of MED28 using si-RNA significantly increased the sub-G1 cell population and decreased the population of the cells in the S phase and at the G2−M transition (Figure 4A,C). Furthermore, the knockdown of MED28 using si-RNA in the HeLa cells stably expressing MED28 significantly increased the sub G1 and G1 cell populations and decreased the populations of cells in the S phase and at the G2−M transition, as well as the population of cells with polyploidy over 4n (Figure 4A,D).

It is known that the enhanced progression of the cell cycle induces aneuploidy [24]. Given that the overexpression of MED28 also increased the cell population with polyploidy of over 4n, as shown in Figure 4A–D and Appendix A, we examined whether the overexpression of MED28 could induce aneuploidy. Immunofluorescence analyses showed that the overexpression of MED28 significantly increased the rate of formation of a micronucleus and nuclear budding, which are hallmarks of chromosomal instability (Figure 5A–C). In addition, the numbers of the binucleated and multinucleated cells, representing aneuploidy, significantly increased when compared to the aneuploidy that was observed in the control and empty-vector groups (Figure 5A,D). Taken together, these results suggest that over-expression of MED28 shortens the cell cycle by promoting G1–S and G2–M transition, inducing chromosomal instability, which may at least, in part, lead to aneuploidy.

## 3. Discussion

Recent studies indicate that the overexpression of *MED28* induces cancer cell proliferation and increases the colony-forming ability of cancer cells upon DNA damage [11,13]. In this study, we first identified four transcription factors, namely E2F-1, NRF-1, ETS-1, and C/EBPβ, which increase the expression level of *MED28*. E2F-1 is reported to be an oncogene that is involved in DNA replication and the G1–S phase transition, and hence affects cell cycle progression. [25]. The overexpression and gene amplification of E2F-1 in a variety of cancers are reported. E2F-1 transgenic mice develop cancers in the vagina, skin, forestomach, and odontogenic epithelium [26]. Moreover, the overexpression of E2F-1 has been reported to have a prognostic value in breast cancer [26]. In addition, the target genes of E2F-1 are known to be required in cell cycle progression and they are mainly involved in DNA synthesis [27]. The screening of MED28 binding proteins in *MED28*-overexpressing cells using liquid chromatography coupled to tandem mass spectrometry (LC-MS/MS) showed that a putative binding protein for MED28 is similarly associated with DNA replication (results not shown). However, this association needs further elucidation. In addition, ETS-1 is mainly expressed in triple-negative breast cancer, which is closely associated with poor survival. Furthermore, ETS-1 induces cancer cell invasion and regulates epithelial-to-mesenchymal transition (EMT), thereby leading to drug resistance and neovascularization [28].

Of note, an in vitro drug resistance model that is based on a breast cancer cell line has been reported to manifest enhanced expression of MED28, which regulates epithelial–mesenchymal transition in human breast cancer [29]. C/EBPβ is associated with metastatic breast cancer and an overall poor prognosis [30,31]. In addition, the high expression of NRF-1 is closely correlated with poor survival in the luminal A (ER^+^/PR^+^/HER2^−^) subtype of breast cancer [32]. Taken together, these observations strongly suggest that the increase in MED28 expression by E2F-1, ETS-1, C/EBPβ, and NRF-1 is associated with cancer progression and poor prognosis.

As E2F-1 is an oncogene that stimulates DNA replication and the G1–S phase transition [25], and MED28 is one of its target genes, we hypothesized that the pattern of MED28 expression would change during cell cycle progression. As illustrated in Figure 2, the expression level of MED28 gradually increased from the early G1 phase until the early S phase and decreased from mid-S phase to mid-G2 phase, and then increased again from the late G2 to M phase. Decreased expression of MED28 was observed as cells exited mitosis. These findings clearly show that MED28 is involved in cell cycle regulation, at least in part.

Cyclin-dependent kinases (CDKs), a family of serine/threonine kinases, control the cell cycle by forming an active heterodimeric complex with cyclins. CDK2 forms a complex with cyclin E for the G1–S transition and with cyclin A for S phase progression [33]. Therefore, we examined whether MED28 contains a motif that is phosphorylated by CDK, being represented by the sequence (S/T)PX(R/K). We found that MED28 had two consensus motifs that were recognized by CDK (Appendix A). An in vitro binding assay using CDKs confirmed that MED28 can bind to CDK2 (Appendix A). In addition, the post-translational modification analysis revealed that residues S103 and T134 of MED28 were phosphorylated in HeLa cells (results are not shown). Retinoblastoma protein (pRb) inhibits the transcriptional activity of E2F-1 by forming the pRb/E2F-1 complex. The active CDK2–cyclin complex hyperphosphorylates pRb, which releases E2F-1 and enables E2F-1 to function as an active transcription factor [34]. Our study showed that MED28 expression is cell cycle dependent and increased by E2F-1, and that CDK2 may mediate the phosphorylation of MED28. Thus, CDK2 and E2F-1 might regulate the increase in MED28 levels at the G1–S transition. However, the underlying molecular mechanism needs further elucidation.

Defects at the cell cycle checkpoints can be induced by alterations of transcriptional or post-translational regulation via tumor suppressor genes or oncogenes [35]. In addition, tumor cells that progress with abnormal chromosome segregation during mitosis acquire aneuploidy. Given that mitotic checkpoints are important for precise chromosome segregation, aberrant cell division that is due to abnormal mitotic checkpoints often induces aneuploidy in cancer cells [35]. In this study, we found that the overexpression of MED28 shortened the mitosis by increasing chromosome condensation, segregation, cytokinesis, and decondensation, which resulted in a more abnormal nuclear phenotype (nuclear budding and the presence of a micronucleus) and aneuploidy (Figure 3, Figure 4 and Figure 5). This phenotype was further confirmed by the FACS analyses, as depicted in Figure 4 and Appendix A. In addition, the overexpression of MED28 decreased cell populations at the G1–S transition. The mechanism of MED28-driven shortening of the S phase and the subsequent induction of chromosome instability and aneuploidy need further elucidation. We could not elucidate the molecular mechanism behind the abnormal cell cycle and aneuploidy induced by MED28 overexpression in this study. However, cancer genomics data, where *MED28* was found to be amplified in various cancers, including breast cancer, ovarian cancer, pancreatic cancer, sarcoma, lung adenoma, prostate cancer, and melanoma, clearly indicate that the overexpression of MED28 is closely related to cancer progression (Appendix A). In addition, the analyses of RNA-seq data from various cancers suggest that the gene amplification of *MED28* closely correlates with an increase in MED28 mRNA expression (Appendix A) [10,36,37].

Overall, we have identified and characterized the transcription factors that increase MED28 expression. In addition, we found that the regulation of MED28 expression is cell cycle dependent, and MED28 overexpression induces abnormal cell cycle progression, genomic instability, and aneuploidy. Therefore, our results suggest that MED28 plays a critical role in tumorigenesis by inducing genomic instability.

## 4. Materials and Methods

### 4.1. Cell Culture and Plasmid Construction

HeLa, HEK293, and MCF7 cells were cultured in Dulbecco’s modified Eagle’s medium (DMEM; Hyclone, GE Healthcare Life Sciences, Marlborough, MA, USA), supplemented with 10% (*v*/*v*) fetal bovine serum (FBS; Hyclone, GE Healthcare Life Sciences, UT, USA), and 1% (*v*/*v*) penicillin/streptomycin solution (PS; Hyclone, GE Healthcare Life Sciences, UT, USA) in a humidified CO_2_ incubator. Full-length MED28, CDKs, ETS-1, E2F-1, NRF-1, and C/EBPβ cDNAs were PCR-amplified and subcloned into vectors pcDNA3-Flag, pcDNA3-Myc (Invitrogen, Carlsbad, CA, USA), pEXPR-Iba-105 (Iba Lifescience, Goettingen, Germany), or pEGFP-C1 (Clontech, Mountain View, CA, USA) vectors.

### 4.2. Mammalian Cell Transfection

Plasmid transfection was performed while using a mixture of 150 mM NaCl and polyethylenimine (PEI; Polysciences, Inc., Warrington, PA, USA) or Lipofectamine™ 2000 (Invitrogen, CA, USA) according to the manufacturer’s protocols.

### 4.3. The Luciferase Reporter Assay

The *MED28* promoter region was analyzed in UCSC Genome Browser (http://genome.ucsc.edu/). The *MED28* promoter region was cloned into the pGL3-basic luciferase reporter vector. The pGL3 basic vector or pGL3 basic vector containing the *MED28* promoter (200 ng) with a *Renilla* vector (100 ng) was transfected into the HEK293 cells. To examine the effects of transcription factors, the HEK293 cells were cotransfected with pGL3-MED28 promoter (200 ng), with *Renilla* vector (100 ng), and one of the transcription factor constructs (each 200 ng, ETS-1, E2F-1, NRF-1, and C/EBPβ) using PEI. After 24 h incubation, the cells were lysed with passive lysis buffer, and a Dual Luciferase Reporter Assay System measured luciferase activity (Promega, Madison, WI, USA) and quantitated using GloMax (Promega, WI, USA) according to the manufacturer’s instructions.

### 4.4. The ChIP Assay

After the transfection of the empty vector or the ETS-1, E2F-1, NRF-1, or C/EBPβ plasmid (5 μg) into HEK293 cells, the cells were fixed with 1% (*w*/*v*) formaldehyde for 10 min at 25 °C. ChIP was performed, as described previously [7]. The cells were incubated in 200 µL of sonication buffer and sonicated for 70 s (10 s pulse and 60 s rest). DNA was sheared to lengths between 200 and 1000 bp. The sonication mixture was incubated with mock immunoglobulin G (Santa Cruz Biotechnology, CA, USA), anti-E2F-1 (Santa Cruz Biotechnology, Dallas, TX, USA), anti-NRF-1 (Santa Cruz Biotechnology, CA, USA), anti-ETS-1 (Santa Cruz Biotechnology, CA, USA), or anti-C/EBP antibody (Santa Cruz Biotechnology, CA, USA), followed by immunoprecipitation using Protein A and G agarose bead mixture (Invitrogen, Paisley, UK). The protein–DNA complexes were separated from the beads by elution buffer (50 mM Tris HCl, pH 8.0, 1 mM EDTA, 1% [*w*/*v*] sodium dodecyl sulfate (SDS), 50 mM NaHCO_3_), and then heated at 65 °C for 5 h with protease K to reverse the formaldehyde crosslinks. Finally, the DNA was purified with the PCR Clean-up Kit (Promega, WI, USA). ChIPed DNA was used as a template for PCR (Forward: 5′-GCCTACTACACCAGCC-3′, Reverse: 5′-CATGTTTGGAATGGCG-3′).

### 4.5. Cell Cycle Analysis

HeLa cells were arrested at the G1–S or G2–M transition by a double thymidine block (2 mM) or thymidine-nocodazole block (thymidine 2 mM, nocodazole 100 ng/mL), respectively. The synchronized cells were harvested, fixed in 70% (*v*/*v*) ice-cold ethanol, washed with ice-cold phosphate-buffered saline (PBS), and then stained with 40 μg/mL propidium iodide (PI) in the presence of 50 μg/mL RNase A for 30 min at 25 °C. All of the DNA samples (1 × 10^4^ cells/sample) were analyzed with a Cube 6 system (Sysmex Partec, Sysmex, GmbH, Germany) and in the FCS Express 4 Flow Research Edition (De Novo Software™, Glendale, CA, USA) software. To generate G2–M-synchronized cell populations, the HeLa cells were synchronized by treatment with thymidine (2 mM) to generate a population enriched in cells in the postmitotic and G1 phases of the cell cycle, and the cells were then released from the arrest for 9 h with a fresh cell culture medium. The cells were then treated with nocodazole (100 ng/mL) for 16 h, followed by a release from the arrest with a fresh cell culture medium. The cells were harvested at various time points after synchronization and then analyzed for DNA content by flow cytometry, as described above.

### 4.6. Immunoblot Analysis, Strep-Tactin-Based Protein Purification, Antibodies, and Chemicals

The cells were lysed with lysis buffer (1% [*v*/*v*] Triton X-100, 50 mM HEPES, pH 7.6, 150 mM NaCl, 10% [*v*/*v*] glycerol, 10 mM NaF, 1 mM Na_3_OV_4_, 1 mM phenylmethylsulfonyl fluoride, and 1 mM EDTA) supplemented with a protease inhibitor cocktail (PIC; Thermo Fisher Scientific, Waltham, MA, USA), and the proteins were subjected to SDS polyacrylamide gel electrophoresis (SDS-PAGE) and then transferred onto polyvinylidene fluoride (PVDF) membrane (Millipore, Burlington, MA, USA), followed by antibody-based detection using an Enhanced Chemiluminescence Eetection system (Santa Cruz Biotechnology, TX, USA). For streptavidin-based protein purification, the HEK283 cells were transfected with the pEXPR-IBA103-MED28 plasmid in the presence of Myc-CDK1, Myc-CDK2, Myc-CDK3, or Myc-CDK4 for 24 h and then lysed with lysis buffer (1% [*v*/*v*] Triton X-100, 50 mM HEPES, pH 7.6, 150 mM NaCl, 10% [*v*/*v*] glycerol, 10 mM NaF, 1 mM Na_3_OV_4_, 1 mM phenylmethylsulfonyl fluoride, and 1 mM EDTA) containing the protease inhibitor cocktail at 4 °C for 30 min. Subsequently, Strep-Tactin Sepharose beads (Iba Lifesciences, Gottingen, Germany) were added and rotated at 4 °C for 2 h. The beads were collected by centrifugation and then washed three times in lysis buffer. The samples were boiled in SDS-PAGE sample buffer at 100 °C for 10 min and resolved by SDS-PAGE on a 12.5% gel and analyzed by immunoblotting. The antibodies used in this study included M2 anti-Flag (Sigma Aldrich, St. Louis, MO, USA), anti-Myc (1:1000; Santa Cruz Biotechnology, Dallas, TX, USA), anti-HA (1:1000; Santa Cruz Biotechnology, TX, USA), anti-MED28 (1:10,000; laboratory-made), and anti-γ-tubulin (1:20,000; laboratory-made).

### 4.7. A Glutathione S Transferase (GST) Pull-Down Assay

GST pull-down assay was performed, as described previously [9]. Briefly, the pGEX4T-1 vector or pGEX4T-1 vector containing MED28 was transformed into the BL21(DE3) *Escherichia coli* strain, and the expression of the GST or GST-MED28 protein was induced while using 0.1 mM IPTG at 30 °C for 5 h. The cells were harvested and then lysed in PBS containing 0.1% (*v*/*v*) Triton X-100 with sonication. The cell extracts containing GST or GST-MED28 were bound to glutathione-Sepharose beads (GE healthcare, Pittsburgh, PA, USA), and the beads were washed five times with PBS containing 0.1% (*v*/*v*) Triton X-100 to remove the nonspecifically bound proteins. Beads containing GST or GST-MED28 were incubated with HEK293 cell lysates containing Myc-CDK1 or Myc-CDK2 for 2 h at 4 °C, washed three times with PBS containing 0.1% (*v*/*v*) Triton X-100, and then boiled in SDS-PAGE sample buffer at 100 °C for 10 min. The samples were subjected to SDS-PAGE for immunoblot.

### 4.8. Live Cell Imaging and Fluorescence Microscopy

The HeLa cells were transfected with the pcDNA3 vector or pcDNA-mycMED28 and selected with G418 (200 μg/mL) for 10 days. For the estimation of mitosis duration and whole cell cycle analysis with time-lapse photomicroscopy, HeLa cells that were transfected with the pcDNA3 vector or pcDNA-Myc-MED28 were seeded (1 × 10^4^ cells/well) in a four-well glass dish (Thermo Scientific™ Nunc™Lab-Tek II Chambered Coverglass, MA, USA) and incubated overnight in standard culture conditions. To visualize chromosomes, the cells were incubated with 1 μg/mL Hoechst 33342 (Thermo Scientific™ Hoechst^®^ 33342, MA, USA) for 30 min. Fluorescence images were acquired every 5 min for 24 h while using a Nikon eclipse Ti camera (Tokyo, Japan) with a 40× dry Plan-Apochromat objective. Images were captured with an iXonEM +897 Electron Multiplying charge-coupled device camera (Teledyne Princeton Instruments, Trenton, NJ, USA) and analyzed in the Nikon Imaging Software (NIS)-elements advanced research (AR) (Nikon, Tokyo, Japan). For immunocytochemistry, the cells were fixed in 4% formaldehyde in PBS for 15 min at 25 °C and permeabilized with 0.2% (*v*/*v*) Triton X-100 (Sigma Aldrich, MO, USA) in PBS for 10 min at 25 °C. The fixed cells were preincubated in blocking solution (3% (*w*/*v*) bovine serum albumin in PBS), followed by incubation with primary antibodies at 4 °C overnight. After incubation with primary antibodies, the cells were washed three times via shaking in PBS and probed with fluorescein (Cy3, Alexa 488)-conjugated anti-mouse or anti-rabbit secondary antibodies. After three washes with PBS, DAPI (1 μg/mL, Invitrogen, CA, USA) was used for DNA counterstaining. The cells were then washed three times with PBS and subsequently incubated in a mounting solution (Biomeda, CA, USA). The samples were examined under a fluorescence (Axio Imager M1, Carl Zeiss, Oberkochen, Germany) or confocal microscope (LSM710 or LSM510, Carl Zeiss, Oberkochen, Germany).

### 4.9. Statistical Analysis

The results are expressed as mean ± SD or mean ± SEM. A student’s *t* test was performed to determine the statistical significance of all the results obtained. *p* < 0.05 was considered to be statistically significant.

## Figures and Tables

**Figure 1 ijms-20-01746-f001:**
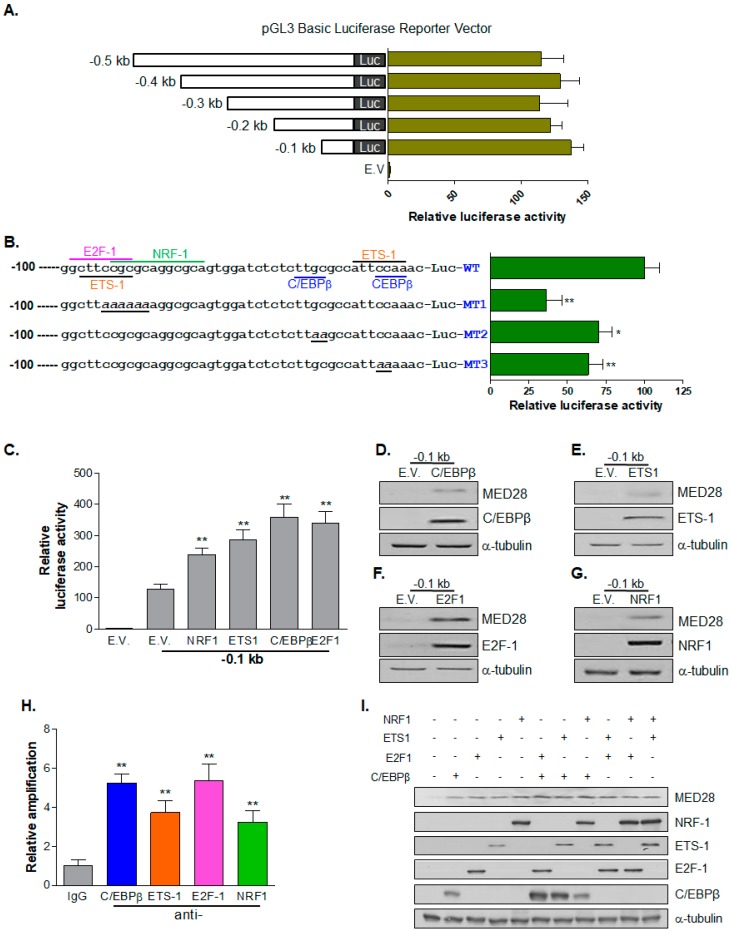
Analysis of the mammalian mediator complex subunit 28 (*MED28)* promoter. (**A**) pGL3-basic vectors containing the indicated version of the *MED28* promoter were transfected into cells, and luciferase activity was measured as described in the Materials and Methods section. The luciferase activity was normalized to *Renilla* luciferase activity. Data represent the mean ± SEM of three independent experiments (*n* = 3). (**B**) The putative transcription factor-binding sites were analyzed and are indicated in the −0.1 kb region of the *MED28* promoter. PGL3-basic vectors containing wild type (WT) or mutant (MT) promoters were transfected into HEK293 cells for 24 h, and the luciferase activity was measured. Data represent the mean ± SEM of three independent experiments (*n* = 4; * and **, vs. WT). (**C**) The effect of E2F-1, NRF-1, ETS-1, and C/EBPβ on the −0.1 kb promoter region of the *MED28* promoter was examined by the luciferase assay. Data represent the mean ± SEM of three independent experiments (*n* = 4; **, vs. empty vector (E.V.)/−0.1 kb). (**D**–**G**) The expression level of MED28 was examined by western blot after transfection of the indicated transcription factors. (**H**) Each vector encoding transcription factor was transfected into HeLa cells for 24 h, and a chromatin immunoprecipitation (ChIP) assay using control (IgG) or each specific antibody was carried out. ChIPed DNA was analyzed by quantitative polymerase chain reaction (qPCR) with *MED28* promoter specific primers between positions −100 and −1 bp as described in the Materials and Methods section. Results are from at least three independent experiments (*n* = 4; **, vs. immunoglobulin G [IgG]). (**I**) After combinatorial transfection of E2F-1, NRF-1, ETS-1, and C/EBPβ (0.5 μg of each vector DNA), the expression level of MED28 was examined by western blot. Tubulin was used as a loading control. * *p* < 0.05, ** *p* < 0.01.

**Figure 2 ijms-20-01746-f002:**
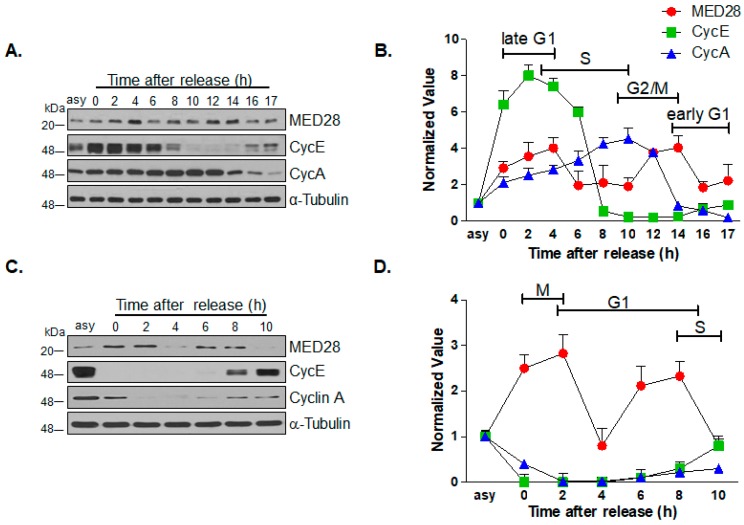
The changes in MED28 expression during cell cycle progression. (**A**) HeLa cells were synchronized at the G1–S transition by double thymidine (2 mM) block and then released from the arrest for the indicated times and analyzed by western blot with the respective antibodies. The expression levels of cyclin A and cyclin E were used as cell cycle markers, and tubulin was used as a loading control. The results are from at least three independent experiments are depicted (*n* = 5). Asy, asynchronized. (**B**) Changes in the expression level of each protein after the release from thymidine block. The results are from at least three independent experiments are presented (*n* = 5). (**C**) HeLa cells were arrested at the G2–M transition by treatment with thymidine (2 mM) and nocodazole (100 ng/mL), and then released from the arrest. The cells were harvested at the indicated time points for western blot. The expression levels of cyclin A and cyclin E were used as cell cycle markers, and tubulin was used as a loading control. Results from at least three independent experiments are depicted (*n* = 5). (**D**) The change in the expression level of each protein after the release from nocodazole block. Results from at least three independent experiments are presented (*n* = 5).

**Figure 3 ijms-20-01746-f003:**
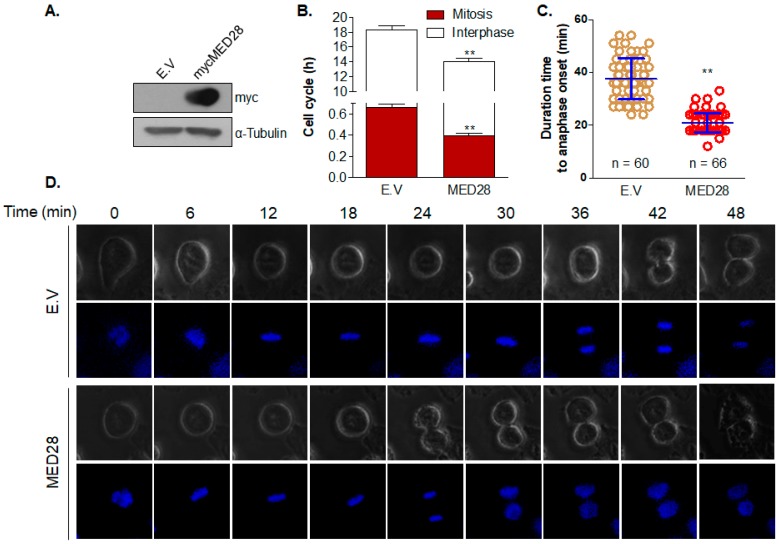
MED28 overexpression promotes cell division. (**A**) After transfection of the control Myc-expressing vector (empty vector; E.V.) or the Myc-MED28 plasmid into HeLa cells, a stable cell line was selected with G418 as described in the Materials and Methods section. MED28 expression was confirmed by western blotting. MED28 was detected with an anti-Myc antibody. Tubulin was used as a loading control. (**B**) The whole cell cycle duration was investigated using time-lapse photomicroscopy as described in the Materials and Methods section. At least 60 cells per sample were counted per experiment, and the results from at least three independent experiments are presented (**, vs. E.V.). (**C**,**D**) HeLa cells harboring control E.V. or stably expressing MED28 were cultured and stained with Hoechst. The period from prophase to anaphase onset, defined as chromosome separation, was measured. The cells were imaged by a time-lapse photomicroscopy during mitotic progression under 60× magnification. Photos were taken every 3 min, and the start time was set to the nuclear envelope breakdown (**, vs. E.V.). ** *p* < 0.01.

**Figure 4 ijms-20-01746-f004:**
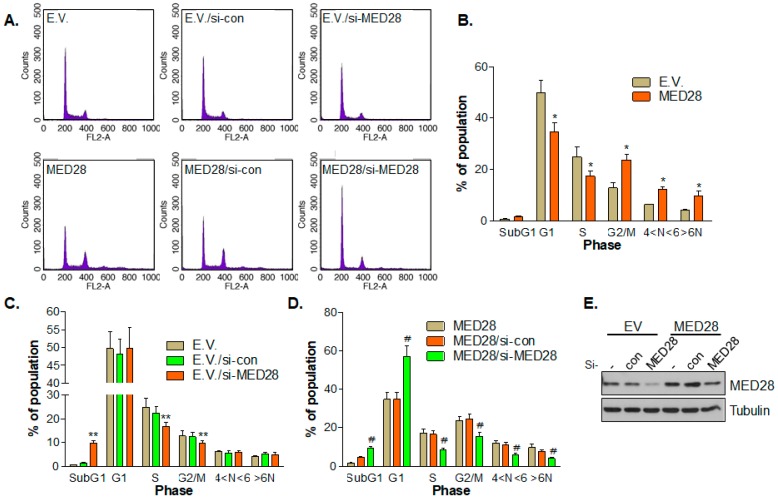
MED28 controls the cell cycle. (**A**) The stable HeLa cell lines harboring empty vector (E.V.) or overexspressing MED28 were transfected with the control or MED28 si-RNA (20 nM) for 48 h and harvested for fluorescence-activated cell sorting (FACS) analysis to compare their DNA contents. Results from at least three independent experiments are presented (*n* = 3). (**B**–**D**) The cell population in each phase was evaluated and presented (*, vs. E.V.-harboring cell population; **, vs. E.V./si-con-harboring cell population; #, vs. MED28/si-con-expressing cell population). Results from at least three independent experiments are depicted (*n* = 3). (**E**) The overexpression or knockdown of MED28 was confirmed by immunoblotting. Tubulin was used as a loading control. The results are depicted as the mean ± SD. * *p* < 0.05, ** *p* < 0.01, and ^#^
*p* < 0.01.

**Figure 5 ijms-20-01746-f005:**
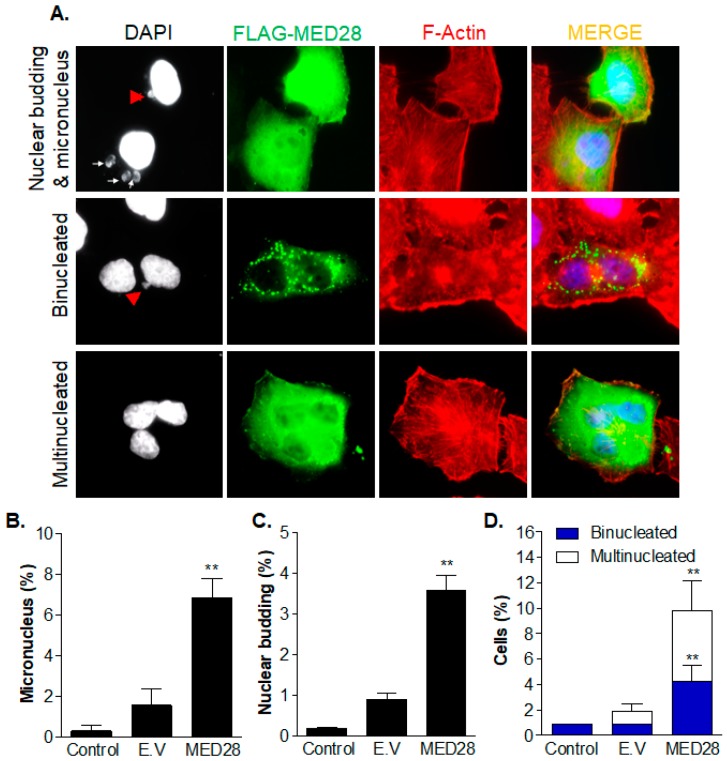
Overexpression of MED28 induces genomic instability and aneuploidy. (**A**) Anti-Flag (green) and anti-F-actin (red) immunostaining of HeLa cells transfected with the Flag-MED28 construct. 4′,6-Diamidino-2-phenylindole (DAPI) was used for nuclear staining. The white arrow and red arrowhead indicate a micronucleus and nuclear budding, respectively. Fluorescence images were captured by confocal microscopy under 60× magnification. (**B**–**D**) The micronuclei, nuclear budding, binucleated cells, and multinucleated cells were counted in control, empty vector (E.V.)-transfected, and MED28-expressing HeLa cells. The results are shown as mean ± SD from at least three independent experiments (*n* = 4) (**, vs. E.V.). ** *p* < 0.01.

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
