# Peer review of "MED28 Over-Expression Shortens the Cell Cycle and Induces Genomic Instability"

_ijms, 2019, doi:10.3390/ijms20071746_

Round 1
Reviewer 1 Report
The authors study the MED28 subunit of mediator and identify several transcription factors that modulate the expression of MED28. The authors then try to correlate the overexpression and knockdown of MED28 with changes in the cell cycle and genomic instability. While the authors present convincing data to support E2F1, NRF1, ETS1, and C/EBPbeta as potential transcriptional regulators of MED28, the studies which overexpress and knockdown MED28 do not convincingly demonstrate a central role for MED28 in modulating the cell cycle directly or in causing genomic instability. The studies on cell cycle regulation are suggestive, but do not go far enough to conclude that MED28 is involved in cell cycle regulation. As an E2F1 target, MED28 likely mediates some of the events of G1/S phase, but the direct role of MED28 in cell cycle regulation is unclear. Additionally, the experiments presented on genomic instability and aneuploidy do not directly show the development of aneuploidy as a consequence of MED28 overexpression. Instead, the authors suggest that the presence of nuclear budding and micronuclei is suggestive of genomic instability, but this is not convincing and these are not traditional markers of aneuploidy or genomic instability. The authors should be careful not to overstate their findings on cell cycle regulation and genomic instability by MED28.
Author Response
Thank you for your kind review of this manuscript. The responses to all comments have been prepared and given below.
Q1. The studies on cell cycle regulation are suggestive, but do not go far enough to conclude that MED28 is involved in cell cycle regulation.
Answer: Thank you for your critical suggestion. As you said, we did not conclude that med28 is involved in cell cycle regulation, but instead moderated the expression in the abstract and in the main text as following “overexpression of MED28 disturbs the cell cycle”.
Q2. As an E2F1 target, MED28 likely mediates some of the events of G1/S phase, but the direct role of MED28 in cell cycle regulation is unclear
Answer: You are correct. We aimed to show the feasibility that MED28 overexpression as an E2F1 target, can disturb the normal cell cycle. Direct role in cell cycle regulation needs further study, which is described in discussion section in detail. In addition, we are currently studying about that. Thank you for your critical suggestion.
Q3. The studies which overexpress and knockdown MED28 do not convincingly demonstrate a central role for MED28 in modulating the cell cycle directly or in causing genomic instability.
Answer: In this study, we could not show the mechanism how MED28 modulates the cell cycle and causes genomic instability. However, we have shown at least that aneuploid cell population as well as cell cycle changes when the expression of MED28 is up-regulated or down-regulated. In addition, we described in discussion as followings “The mechanism of MED28-driven shortening of the S phase and subsequent induction of chromosome instability and aneuploidy need further elucidation. We could not elucidate the molecular mechanism behind the abnormal cell cycle and aneuploidy induced by MED28 overexpression in this study. Cancer genomics data, however, where MED28 was found to be amplified in various cancers, including breast cancer, ovarian cancer, pancreatic cancer, sarcoma, lung adenoma, prostate cancer, and melanoma, clearly indicate that the overexpression of MED28 is closely related to cancer progression (Supplementary Fig. 4A). In addition, the analyses of RNA-seq data from various cancers suggest that gene amplification of MED28 closely correlates with an increase in MED28 mRNA expression (Supplementary Fig. 4B)”. As you suggested, this study needs further study to reveal the mechanism in detail.
Q4. The authors should be careful not to overstate their findings on cell cycle regulation and genomic instability by MED28.
Answer: Thank you for your comment. We have moderated the expression about the results as much as possible, as you suggested.
In addition, we have corrected this manuscript again by native English editor.

Reviewer 2 Report
In the given manuscript, Cho et al have identified transcription factors,including E2F-1, NRF-1, ETS-1, and C/EBPb, as transcriptional regulators of MED28 using Luciferase-reporter and CHIP assays. MEDD28 was shown to be regulated in a cell-cycle dependent manner. Moreover, overexpression of MEDD28 induced cell cycle abnormalitites and aneuploidy and micronuclei. Overall, the findings are novel. The manuscript is very well written and experimental design is good. Data quality is very good , and supports conclusions in most part.
Author Response
Thank you for your kind review!
Reviewer 3 Report
In the present manuscript, authors identified the transcription factors that increase MED28 expression by luciferase reporter assay. In addition, the authors observed an enhanced MED28 expression in the G1−S phase and during mitosis. Furthermore, over-expression of MED28 significantly decreased the duration of interphase and mitosis. Conversely, knockdown of MED28 using si-RNA increased the duration of interphase and mitosis. Interestingly, over-expression of MED28 significantly increased micronucleus and nuclear budding in HeLa cells. In addition, flow cytometry and fluorescent microscopy analyses showed that over-expression of MED28 significantly increased aneuploid cells.
Overall, I think that the experimennts performed, supported the conclusions of authors.
Author Response
Thank you for your kind review!
In addition, we have corrected this manusccipt again by native English editor.